# Performance Analysis of Sphere Packed Aided Differential Space-Time Spreading with Iterative Source-Channel Detection

**DOI:** 10.3390/s21165461

**Published:** 2021-08-13

**Authors:** Hameed Ullah Khan, Nasru Minallah, Arbab Masood, Amaad Khalil, Jaroslav Frnda, Jan Nedoma

**Affiliations:** 1Department of Computer Systems Engineering, University of Engineering and Technology Peshawar, Peshawar 25000, Pakistan; hameedkhan.cse@uetpeshawar.edu.pk (H.U.K.); n.minallah@uetpeshawar.edu.pk (N.M.); arbabmasood@uetpeshawar.edu.pk (A.M.); 2Department of Quantitative Methods and Economic Informatics, Faculty of Operation and Economics of Transport and Communication, University of Zilina, 01026 Zilina, Slovakia; jaroslav.frnda@fpedas.uniza.sk; 3Department of Telecommunications, Faculty of Electrical Engineering and Computer Science, VSB-Technical University of Ostrava, 70800 Ostrava, Czech Republic; jan.nedoma@vsb.cz

**Keywords:** sphere packing, differential space time spreading, bit error ratio (BER), EXtrinsic Information Charts (EXIT), H.264

## Abstract

The introduction of 5G with excessively high speeds and ever-advancing cellular device capabilities has increased the demand for high data rate wireless multimedia communication. Data compression, transmission robustness and error resilience are introduced to meet the increased demands of high data rates of today. An innovative approach is to come up with a unique setup of source bit codes (SBCs) that ensure the convergence and joint source-channel coding (JSCC) correspondingly results in lower bit error ratio (BER). The soft-bit assisted source and channel codes are optimized jointly for optimum convergence. Source bit codes assisted by iterative detection are used with a rate-1 precoder for performance evaluation of the above mentioned scheme of transmitting sata-partitioned (DP) H.264/AVC frames from source through a narrowband correlated Rayleigh fading channel. A novel approach of using sphere packing (SP) modulation aided differential space time spreading (DSTS) in combination with SBC is designed for the video transmission to cope with channel fading. Furthermore, the effects of SBC with different hamming distances $d_{\left(H,min\right)}$ but similar coding rates is explored on objective video quality such as peak signal to noise ratio (PSNR) and also the overall bit error ratio (BER). EXtrinsic Information Transfer Charts (EXIT) are used for analysis of the convergence behavior of SBC and its iterative scheme. Specifically, the experiments exhibit that the proposed scheme of error protection of SBC $d_{\left(H,min\right)}$ = 6 outperforms the SBCs having same code rate, but with $d_{\left(H,min\right)}$ = 3 by 3 dB with PSNR degradation of 1 dB. Furthermore, simulation results show that a gain of 27 dB $E_{b}/N_{0}$ is achieved with SBC having code rate 1/3 compared to the benchmark Rate-1 SBC codes.

## 1. Introduction

Faced with the increasing demands of enhanced connectivity with higher data rates, and in order to ensure the provision of reliable communication, the multimedia broadcasting research community has undertaken advanced research in promoting beyond 4G wireless communication [1]. Over wireless communication, the significance of multimedia communication cannot be overstated. However, the wireless transmission is restricted by limited data rates, in the form of transmission power and limited available bandwidth [2]. According to a report of Cisco, by 2023, there would be 5.3 billion global internet users, with half of the devices supporting mobile video streaming [3]. Furthermore, the report estimates that with the evolution of 5G, the traditional data speed would enhance up to 13 times. 5G is in the process of implementation, and global preparations are made to adopt to the new standards, and expectations are very high. 5G is expected to handle Ultra-High-Definition (UHD) video for the wireless streaming services, and at the same time it is believed that the quality of experience (QoE) would meet the end user requirements and network conditions [4]. Many new features are being incorporated in 5G such as ultra-high speed internet, online gaming, multi-person video conferencing and voice communication, 3D and UHD video, smart grid, and a unified global standard for communication [5]. Multimedia communication has received overwhelming attention in its transmission over the wireless channels [6,7]. To overcome these challenges of band-limited channels and increased data size, the data to be sent is compressed using various data compression standards [8,9]. The compression standards consider the approach of removing redundancy from the transmission stream. However, this makes the compressed data bitstream prone to transmission errors [10]. Thus, there is an inevitable need of such a scheme to maintain optimum integrity while accommodating maximum compression [8]. Traditionally, the source and channel coding were separately optimized but the joint source and channel decoding (JSCD) helped achieve better optimization and performance, therefore attracting much attention [11,12,13,14,15]. The JSCD [16] technique cashes in on the leftover redundancy in the precoded source data bitstream and exploits it for the error protection, which makes the JSCD of prime interest for wireless transmission [17,18]. Fingscheidt and Vary [19] proposed a soft bit source decoding (SBSD) scheme for exploitation of the residual bitstream redundancy and exhibited substantial improvement in the convergence behavior exhibited by the iterative source-channel decoding (ISCD) [18]. Iterative source and channel decoding (ISCD) aims to assist the inner and outer decoders in an iterative manner to find the maximum possible extrinsic information. However, the use of an advanced coding scheme only leaves behind very limited residual redundancy. Therefore, this study proposes further deliberate redundancy addition through novel source bit coding (SBC). In SBC, artificial redundancy is added in the source coded bitstream to attain better performance of iterative decoding. In this research, H.264/AVC [20] with variable predictive coding techniques is used to increase its achievable compressional efficiency. This exposes the bitstream to transmission errors [8]. Compressed bitstreams have the shortcoming of the reflection and exaggeration of even a single corrupt bit throughout the transmitted bitstream [21]. Adding on to this, the predictive coding may propagate the error to the neighbor frames and neighboring blocks of video. Collectively, all these challenges make wireless video transmission very cumbersome [22]. An error resilient data partitioning (DP) [20] scheme has been used to alleviate the data transmission errors. The DP scheme yields three streams with varying coding parameters and importance. Many schemes have been proposed in [8] with improved error resilience, but with the trade-off in computational complexity and reduced compression efficiency. The study [12] presents a serially concatenated code inspired iterative joint source-channel decoding procedure. In [14], a soft-input a posteriori probability decoder based on soft-input was used for exploitation of the residual inherent redundancy for error protection improvement. A novel, near capacity, irregular variable length coding technique was employed in [11] for joint source and channel coding. The authors in [7] put forward advanced design principles of a burst-by-burst adaptive transceiver and its performance analysis for video telephony over cordless and cellular networks. Authors in [15] proposes a MAP algorithm-based JSCD that performs better than previous proposed systems.

In backdrop of the arguments and facts aforementioned, we propose a novel SBC design and defend it with an iterative SBSD scheme ensuring convergence and exhibiting an $E_{b}/N_{0}$ gain of 27 dB at the PSNR degradation point of 1 dB.

The motivation behind this study is to provide reliability and robustness to the wireless transmission of multimedia. The wireless cellular communication is carried out through fast fading channels. However, the shortfall of fast fading channels is that the training sequences are quickly outdated. To overcome this, more training sequences need to be sent, but as a result of this, energy consumption and system overhead is increased. To overcome these effects, we use differential space time spreading (DSTS), which addresses the issue of quickly expiring symbols and also mitigates the interference created by multi-path signals. Furthermore, differential coding also does not need the channel state information (CSI), therefore, our proposed system is independent of CSI. According to [23], wireless transmission experiences diffraction, reflection and scattering, which produce a multipath effect by splitting the signal into multiple copies. The Rayleigh fading channel is considered beneficial for channels with multipath effects, and so in our study, we have used the correlated Rayleigh fading channel. The SP modulation scheme is specially designed for the correlated Rayleigh fading channel, and we have simulated our proposed model for the SP aided DSTS transmission scheme. Secondly, the different code rates of the channel coding techniques have varying effects on the results, which are scrutinized in [24,25,26]. Furthermore, the effects of varying channel coding rates in combination with the differential transmission medium coding and non-typical modulation technique of SP is investigated. By varying the inner and outer rates of the proposed iterative system, the convergence pattern also varies. The EXIT chart analysis validates our proposed system against the benchmarker as it converges at a considerably lower value of $E_{b}/N_{0}$.

The structure of the paper is such that Section 2 covers the related contemporary research work cited and discussed. Section 3 is about source coding technique used (H.264/AVC) while Section 4 is about transmission mediums used in this study. Section 5 is about iterative source-channel decoding and Section 6 is about the source bit coding. In Section 7, EXIT chart analysis is done, and in Section 8, results and performances are discussed. Finally, in Section 9, the conclusion of the study is presented.

## 2. Related Contemporary Work

The compressed video sequences are transmitted over wireless channel encounter transmission errors, which can be overcome by introducing redundancy through certain controlled methodology at the transmitter side while utilizing it at the receiver end for error detection and correction. The author in [27] proposed a video transmission system with different code rates of artificial redundancy with recursive systematic convolutional (RSC) codes having overall half code rate. The author employed SP modulation along with a DSTS scheme that performs better in terms of BER and PSNR results. The authors in [24] compare the performance of three different types of convolutional codes, namely, non-convergent serial concatenated coding, self-concatenated convolutional coding, and convergent serial concatenated coding. The authors established that the self-concatenated convolutional codes give the best possible results. The study [28] puts forward a scheme for coding source and channel jointly to be transmitted through AWGN using neural network algorithms. The proposed system is trained with around 3000 samples. This study comes up with an auto-encoder for a source of *m* dimension to be transmitted over a source with *k* dimension, such that *m > k*. The edge of this study is to set the parameters value randomly, which makes it learn the encoding schemes and helps reach optimum value, so as to find out optimum novice encoders and decoders. In [29], the authors proposed an algorithm for the betterment of JSCC by going deep and also using the neural network algorithms. The research work concluded that deep JSCC performs better than the traditional separation based algorithms. In [30], the technique of JSCC is employed rather than the typical approach of a separate source-channel (SSC) coding scheme. An innovative Bayesian message-exchanging technique is developed, which makes it possible for the units to communicate and the nodes to send data directly instead of decoding each one individually. Performance of the proposed scheme is evaluated by comparing the BER performances of both the SSC and JSCC with various parameters, which shows that the JSCC performs much better. The research work in [31] addresses the issue faced by wireless transmission of digital images by the transmission having the possibility of providing feedback. The additional feature introduced in this research was that of feedback in the AWGN channel. In [32], the granularity in the optic fiber spectral efficiency is evident. The paper claims to have used JSCC for the data transmission through optical fiber over quadrature amplitude modulation (QAM). The hierarchical distribution matching is used with the JSCC. The results show that the peak signal-to-noise ratio (PSNR) performance improvement requires ten times lower $E_{b}/N_{0}$ than the conventional schemes. In [33], superposition coded modulation is used for the video transmission over the wireless channel. The low density parity check (LDPC) codes are used for video encoding, which help in receiving videos of better quality. Here, a scheme is proposed to limit the complexity by iterative decoding. The results exhibit 67% decoding efficiency. In the proposed work [34], the idea behind cooperation is that of apparently using the same medium, but virtually creating a multicast system, and that is done by path sharing. The soft-in and soft-out information of the LDPC codes are exploited to gain both diversity and less-complexity. The codewords are sent by many nodes through a mutual path. The randomness of LDPC codes makes the use of inter-leaver redundant. The exchange of external and internal information by the codes of LDPC makes the cooperation possible and advocates that the cooperative decoding is much better than otherwise. In study [35], JSCC is used along with LDPC for better efficiency. Both the source and channel have similar LDPC codes. For iterative decoding, both of these source and channel codings are to be tanner-graph mapped by using the technique of the message. Three different forms of LDPC codes are simulated in the study. From the results of this study, it is concluded that performance depends on the position of relay. Moreover, it also suggests that the value adaptation of relay and code rates of channel codes can enhance the performance of the system. In [27], the author proposed video transmission system using SP aided DSTS scheme incorporated with SBC scheme with Hamming distance $d_{\left(H,min\right)}$ = 1, while in comparison, our proposed system uses higher $d_{\left(H,min\right)}$, resulting in better performance. Similarly, it outperforms other contemporary studies, such as [24,25,26].

## 3. Source Coding Scheme (H.264/AVC)

The growing demand of high data rate multimedia communication has shifted the burden on the compression efficiency to keep up with the demands of the consumer. Various multimedia communication applications have their specific attributes in the form of error resilience, video quality, compressional efficiency, system complexity, etc. H.264/AVC video codec is currently one of the best solutions out of the globally used video codecs. The H.264/AVC was introduced jointly by the International Organization for Standardization (ISO) and ITU-T Video-coding Experts Group (VCEG). It has, overall, a 94% share of the global usage in all multimedia communication services. This success and global acceptance of H.264/AVC is due to its better transmission robustness, better error resilience and lower complexity as compared to the prevailing codecs.

H.264 is a contemporary video compression standard that is widely implemented in many video applications and is an industry standard. H.264 in incorporated in almost every digital application of video compression, and subsequently, HD television transmission, NETFLIX, YouTube or DVDs all have adopted H.264 as the basic video compression technique. Furthermore, H.264 is not just a part of our daily household, as it has been widely used in the medical field for high quality imagery, and even businesses.

The technology is evolving continuously, and so are the contemporary standards. It is expected that by 2025, most of the conventional HD video transmission systems would be replaced by IP-based systems employing H.264 as the compression standard. H.264 is being extensively studied and modified for additional characteristics such as security [36], latency improvements, and wireless transmission energy consumption for handheld devices [37]. Hence, the importance of H.264 cannot be emphasized any further.

## 4. Transmission Medium Schemes

MIMO: Data sent over the wireless communication networks is prone to attenuation and experience losses during the transmission. Different techniques and settings of the wireless channel units are adopted to overcome these losses. The transmission medium used in this research is that of the Multiple-Input Multiple-Output (MIMO) system. This MIMO system comprises of multiple antennas for sending of multiple copies of the signal from the source side to cope with this attenuation and signal losses. Hence, some of the signals may experience less attenuation as compared to the others. This gives our system the required spatial diversity for improved wireless transmission.DSTS: The Space-Time Spreading (STS) is a technique used to overcome the shortfalls of the wireless channel such as fading of the signals. This technique is used to achieve a higher diversity gain as well as power gain from the transmit signal in multi-user MIMO systems. The advantage of DSTS is that it substantially decreases the complexity of the MIMO channel by eliminating the cumbersome task of channel estimation by employing non-coherent detection method. It provides diversity to wireless channel by employing both spatial and temporal domain coding. This introduces correlation between the signals transmitted in the same time slots. The benefit of space-time coding is that it gives diversity and power gain to the transmitted signal without a need for expansion of the bandwidth with a decreased complexity.

The typical DSTS encoder has an initial block of differential encoder and subsequently an STS for the signal spreading. An example of a differential encoded symbol is [27]:(1)$$v_{t}=\frac{\left(x_{1}\times v_{t-1}^{1}+x_{2}\times v_{t-1}^{2*}\right)}{\sqrt{\left({\left|v_{t-1}^{1}\right|}^{2}+{\left|v_{t-1}^{2}\right|}^{2}\right)}}$$

Similarly, an example of STS of a symbol is:(2)$$c^{T}=\left[\begin{array}{cc} c & c \end{array}\right]$$

Finally the completed DSTS equation for the symbols is:(3)$$y_{t}=c^{T}\times v_{t}$$

SP Modulation: The novelty of this research is the use of SP and orthogonal codes in combination. The use of SP helps in achieving the maximum possible intra-symbol Euclidean distance while keeping the complexity low. The advantage of the combination of SP to DSTS is seen in the form of decreased complexity at the channel level by removing the channel estimation from our system. The reason behind this advantage is the reliance of our proposed system on non-coherent detection of our MIMO system. Furthermore, our proposed system decreases the complexity and cost of the receiver substantially, meanwhile increasing the error resilience. Equation (Equation 4) shows a generic two antenna design on the said principles:

(4)$$G_{2}\left(x_{1},x_{2}\right)=\left[\begin{array}{cc} x_{1} & x_{2}\\ x_{2}^{*} & x_{1}^{*} \end{array}\right]$$

Here, the column represents the spatial dimension and the rows represent the temporal dimension.

## 5. Iterative Source-Channel Decoding

### 5.1. Overview of Proposed System

The organization of ISCD video telephony scheme that we proposed to use in our system for the SBCs performance quantification is shown Figure 1.

The video at the source is compressed for transmission using the H.264 video codec, which yields a bitstream $x_{k}$. This stream is encoded to be mapped into $x_{m}^{{}^{\prime }}$ symbols applying the SBC scheme. Further, the encoded bitstream $x_{m}^{{}^{\prime }}$ is passed through the interleaver η as shown in the Figure 1, generating interleaved bitstream $x_{m}$. Subsequently, this bitstream is encoded using the rate-1 precoder before it is being passed on to the next phase of sphere packing modulation. The sphere packed modulated signal is denoted by $S_{i}$. The DSTS is the final stage on the transmission side. The signal is sent through two antennas while the receiver has a single antenna.

Figure 2 shows the iterative decoder whose statistical independence is given by the EXIT charts, and it employs that it is always related to its length [7]. As a result, rather than performing the ISCD independently on the various frames slices, the bits generated by the MBs are concatenated in single stream and processed so as to achieve improvement in performance without giving in the form of video delay. The precoded $y_{i}$ bitstream ready for transmission is modulated by employing SP for transmission using DSTS. This transmission channel is normalized with Doppler frequency $f_{d}=f_{D}T_{s}=0.01$, where $T_{s}$ is time duration of the signal and $f_{D}$ is Doppler frequency. The soft information obtained at the receiver end from the DSTS modulation is fed into the iterative inner decoder as shown in Figure 1. This step makes it possible to attain the lowest possible BER [13].

### 5.2. The Soft-Bit Source Decoding

The use of soft-bit source decoding (SBSD) is inferred from [13] to obtain the extrinsic log-likelihood ratios (LLRs) for the zeroth Markov model. Furthermore, in our coding scheme, the input bitstream of source is segregated into $M=2^{k}-ary$, or K-bit symbols. This partitioned bitstream is to be encoded by SBCs that we propose. The source bitstream is analyzed and its redundancy is characterized by non-uniform $M=2^{k}-ary$ symbol probability distribution $P[S_{K}\left(t\right)]$, here $S_{K}\left(t\right)=\left[S_{K}\left(1\right),S_{K}\left(2\right),\ldots S_{K}\left(M\right)\right]$. Here, *K* is the total bits in each *M* symbol. As a ground for our calculations, the $M=2^{k}-ary$ bits may be treated as independent of one another. The output information of the symbol *t*th K-bit generated is provided by each single-bit probability product as:(5)$$P\left[y_{\left(T\right)}|y_{T}\right]=\Pi_{k=1}^{K}P\left[{\hat{y}}_{\left(T\right)}\left(k\right)|y_{T}\left(k\right)\right]$$

Here, $\hat{y_{t}}$ are transmitted K-bits and $y_{t}$ are the received K-bit source sequences. The extrinsic output channel information $P[y_{t}]$ is connoted as:(6)$$P\left[{\hat{y}}_{\left(T\right)}^{\left[ext\right]}|y_{\left(T\right)}^{\left[\mathrm{ext}\right]}\right]=\prod_{k=0,k\pm \lambda }^{K}P\left[{\hat{y}}_{\left(T\right)}\left(k\right)|y_{\left(T\right)}\left(k\right)\right]$$

Decisively, LLR as the resultant value for each bit can possibly be obtained by combination of its a priori knowledge and information of channel output of the corresponding symbol [12,13]:(7)$$LLR\left[y_{T}\left(\lambda \right)\right]=\log\left[\frac{\Sigma_{y_{T}\left[ext\right]}p\left(y_{T}\left[ext\right]|y_{T}\left(\lambda \right)=+1\right)\cdot p\left[{\hat{y}}_{\left(T\right)}^{\left[ext\right]}|y_{\left(T\right)}^{\left[ext\right]}\right]}{\sum_{yT^{\left[\mathrm{ext}\right]}}p\left(y_{T}^{\left[ext\right]}|y_{T}\left(\lambda \right)=-1\right)\cdot P\left[{\hat{y}}_{\left(T\right)}^{\left[ext\right]}|y_{\left(T\right)}^{\left[ext\right]}\right]}\right]$$

The SBC scheme proposed in this research has the characteristic of being generic, so it can be applied to any speech or audio and video codecs. Nonetheless, we symbolize in our design the redundancy of the coded bitstream with M-ary, which is a non-uniform symbol probability distribution. For this M-ary we used the 300 frames H.264/AVC video “Akiyo” encoded bitstream, “Miss America”, which has 150-frames and “Mother & Daughter”, which has 300-frames for testing of our system model.

## 6. Source Bit Code Based Iterative Source Channel Decoding

### 6.1. Iterative Convergence Criterion

The ISCD is employed for the purpose of giving the inner and outer decoder the ability to assist each other in order to gain maximum benefit in form of extrinsic information. The performance achievable by the SBSD is constrained by the residual redundancy or the correlation among the coded bits $x_{i}$ because it constrains its acquirable iteration gain. The attainable improvement in performance achievement of SBSD can still be limited despite the use of lossy compression, limited compression and limited delay, which normally would cause typical residual redundancy. In further explanation of this, lower bit-rate video would yield lesser extrinsic information from the precoded source bitstream. The simulation [17] and its results expose that using typical SBSD code provides imperceptible improvement in performance of the system from two iterations onwards. Therefore, we deliberately add artificial redundancy into the bitstream to achieve performance gain through ISCD. This process is done using a novel designed technique referred further on as SBC technique. The innovation in SBC is its exploitation of a distinct property of EXIT chart. Moreover, a receiver with aid of iterative decoder helps achieve near capacity at miniscule BER, if the outer and inner compositional decoder has an open tunnel between their EXIT curves. Kliewer et al. [15] discussed the mandatory and acceptable value of rightful codewords should be $d_{\left(H,min\right)}=2$, for the criterion and quantum of convergence to be met. Furthermore, the perfect input a priori information $H\left(X\right)=L_{\left(SBSD\right)}^{apri}=1$ bit, from the SBSD helps ISCD attain the maximum attainable source entropy $H\left(X\right)=L_{\left(SBSD\right)}^{extr}=1$ bit, if $d_{\left(H,min\right)}=2$. Thus, it motivated us to come up with the novel SBC technique as it is evident that the use of our design scheme all acceptable SBC with an explicit coderate corresponding to the classic coderate, which satisfies the original condition of $d_{\left(H,min\right)}=2$.

### 6.2. Source Bit Codes Algorithm

The sufficient and necessary [13] condition is the foundation for our novel $SBC_{\left[K,N\right]}$ algorithm, explained above, for perfect convergence attainment to negligibly low BER. Our novel $SBC_{\left[K,N\right]}$ segregates the stream $x_{k}$ in K-bit source symbols, to be further encoded as N = (K + P) –bits. Here, P is the count of redundant bits in each K-bit word. We hereby propose $SBC_{\left[K,N\right]}$ encoding scheme for P = (*m*× K) where m>−1. This exhibits a steady increase in $d_{\left(H,min\right)}$ of the coded symbols subsequent to an increase in N and K, provided that the coderate is fixed. The I=[(m−1)XK] number of redundant bits $r_{T(i)}$ are concatenated for *i* = 1, 2, …, I using the K to N–bit method of encoding. This yields [(m−1)XK] bits in total, and the final batch of redundant K bits $r_{T}\left(k\right)$ is yielded for *k* = 1, 2, …, K, by XOR computation of these K bits $b_{T(j)}$, whereas $b_{T}\left[j=k\right]$ is set to 0. This gives us:(8)$$\begin{array}{cc} \; & r_{T}\left(k\right)=\left[b_{T}\left(1\right)⊕b_{T}\left(2\right)\ldots ⊕b_{T}\left(\;K\right)\right];\mathrm{for}\;k=1,2,\cdots ,K\\ \; & \;\mathrm{wbile}\;\mathrm{setting}\;b_{T}\left(k\right)=0 \end{array}$$

Here, ⊕ is a symbol for XOR. To ensure that the total of N-bit resultant codewords demonstrates the $d_{\left(H,min\right)}>2$ among *M* = 2*k*, *k*-bit code words of the source, we infringe meticulously controlled redundancy of the rate of *r* = [*k*/*n*] $SBC_{\left[K,N\right]}$.

### 6.3. Example of Proposed System

For the design part, we illustrate the use and strength of SBCs with a design instance of the scheme. For instance, $SBC_{\left[K,N\right]}$ encoded symbols are yielded by using our novel algorithm with varying rates as rate-1/3, with reciprocal $d_{\left(H,min\right)}$. This is shown in Table 1 with the exclusivity of adding the $r_{T}\left(k\right)$ redundant bits on the right side of *t*-th k bits for *k* = 1, 2, …, K. Table 1 establishes that only 2K of the total possible 2N bit symbols are licit in the mapped coded bitstream. This subsequently establishes that the N-bit has a non-uniform probability.

Table 2 shows the parameters for the coding of the various SBC schemes, which we employ in our novel design instance. Table 2 exhibits that we have maintained a general rate of 1/3 for the SBCs from Table 1, by the use of Rate-1 concatenated precoder, which adjusts to various Rate-1/3 SBC. Furthermore, a concatenated Rate 1/3 RSC is employed for the inner code of Rate-1 SBC, to accomplish the iteration gain, keeping constant the overall bit-rate.

## 7. EXIT Chart Analysis

The EXIT chart of the proposed H.264/AVC coded video transmission system as shown in Figure 2 containing different SBC coding rates, discussed in Table 2 along with Rate-1 Precoder, is shown in Figure 3.

The curves of the EXIT chart in Figure 3 elaborates that the inner Rate-1 SBC scheme curve failed to intersect at the top corner of $\left(I_{A},I_{E}\right)=\left(1,1\right)$. This shows that the inner Rate-1 SBC scheme fails to achieve better transmission and hence gives low BER as compared to other SBC coding rates. The rest of the SBC schemes converge at $\left(I_{A},I_{E}\right)=\left(1,1\right)$, hence it is established that they give better performance as compared to Rate-1 SBC scheme. Different Error Protection code rate schemes employed for outer code SBC with inner precoder of Rate-1 are shown in Table 2. The decoding trajectories of these schemes are shown in Figure 4a–d with $E_{b}/N_{0}=-1$ dB and $E_{b}/N_{0}=-1.5$ dB. Monte Carlo simulations were used for SBSD algorithm in which decoding trajectories were recorded to attain the mutual information for the inner and outer decoder at the input and output. The EXIT chart decoding trajectories between the outer SBC code and the inner Rate-1 Precoder depends upon the $d_{\left(H,min\right)}$ as evident from Figure 4a–d. Figure 4d that advocates that the decoding trajectories for SBC 5/15 gives better performance as it converges near the top corner of EXIT chart on $\left(I_{A},I_{E}\right)=\left(1,1\right)$, as compared to the rest of the coding rates of SBC. The SBC 2/6 having least $d_{\left(H,min\right)}$ as compared to other SBC schemes as shown in Table 2 gives the worst performance of all.

## 8. System Performance and Results

Our proposed system’s performance and results are discussed in this section. Our system contains “Akiyo” video sequence and is represented in QCIF video format. The simulation uses a 45 frames video sequence with each (176 × 144) pixel frame size. Our test sequence consists of a H.264/AVC video codec for encoding with a bit rate of 64 kbps and frame rate 15 frames-per second (fps). Each frame is subdivided into 9 slices, and each slice consists of 11 MBs. Error propagation in frames is reduced by placing each ‘I’ or intra-coded frame followed by 44 ‘P’ or predicted frame with 3 s difference at 15 fps between two consecutive ‘I’ frames. The designed system minimizes the effects of error propagation by using error resilience feature such as data partitioning (DP) and intra-frame coded MBs per frame. Table 3 shows the system parameters of our designed setup.

During our setup for video transmission different error resilient encoding methods were turned off. For inter frame motion compensation, multiple reference frames were deployed, which increased the computational complexity of the system. Flexible macro-block ordering technique complicates the overall system complexity and hence is not used. Both these methods are turned off specially in low motion head and shoulders of different slices of “Akiyo” clip to improve the performance of designed system. In our proposed system for motion search, preceding frame is used instead of using multiple reference frame which supports in reducing the computational complexity. Source encoded bitstream consists of limited residual redundancy so for reducing the computational complexity using no SBC or Rate-1 SBC scheme by decreasing the number of iterations to $I_{t}=5$ between SBSD and RSC decoders. In our simulation, we consider 10 iterations for the SBC scheme having a rate below unity. In our case, the number of iterations is set to 10 because our system is able to attain convergence at 10 iterations. If we further increase the number of iterations beyond 10, there is no significant improvement in the overall performance of the system, but would otherwise add to the delay in the system. Simulation results are generated by taking the average of repeating the experiment 160 times for all 45 frames to achieve more confident results. Performance of different SBC schemes having a Rate-1/3 error protection scheme based on different achievable BER is presented in Figure 4a and described in Table 2. This proposed system is compared with no SBC or with SBC scheme using rate-1 and is portrayed in Figure 4c. Performance analysis in terms of PSNR verses $E_{b}/N_{0}$ graphs are offered in Figure 5b,d and portrays the SBC scheme with $d_{\left(H,min\right)}$ = 6, which gives the best performance in terms of PSNR against $E_{b}/N_{0}$ compared to the available five SBC schemes discussed in Table 2. Figure 5d also shows that the SBSD scheme with a combination of Rate-1/3 inner RSC scheme and Rate-1 outer SBC scheme gives worse PSNR results as compared to the system in which the Rate-1 inner precoder is combined with the 1/3 outer SBC scheme with same overall code rate, as discussed in Table 2. Quantitatively, using the Rate-1/3 SBCs of Table 2, an additional $E_{b}/N_{0}$ gain of up to 27 dB may be achieved over the identical-rate benchmarker scheme dispensing with SBC.

## 9. Conclusions

Through this research paper, we put forward a novel low-complexity SBC technique with spatial diversity integration, which is generic rather than specific. This proposed scheme ensures the essential criteria of $d_{\left(H,min\right)}=2$. We have employed an iterative decoding algorithm in the proposed system for decoding the source bit encoded bitstream for attaining maximum convergence. We have also incorporated various error prevention techniques in our proposed scheme, for the Rayleigh faded transmission of H.264/AVC video. It is meticulously decided to employ SBCs with assorted values of $d_{\left(H,min\right)}$ with the help of various performance evaluation techniques and graphs. This study revealed that bit-error correctional capacity of JSCD was considerably enhanced with the introduction of the SBC codes. This enhancement was achieved due to the conscious induction of redundancy to the compressed source bitstream. Our proposed system introduced spatial diversity with the use of SP modulation and DSTS and hence enhanced the robustness of our transmission system. The use of MIMO made our system state of the art for the use in real world mobile communication system. EXIT charts were produced to analyze the convergence behavior of different code rates of SBCs through the information of the inner and outer decoder of the iterative decoding block. A gain of 3 dB was recorded in the $E_{b}/N_{0}$ at the PSNR degradation point of 1 dB for the H.264/AVC source coded bitstream, while employing $d_{\left(H,min\right)}$ of 6 instead of 1. Moreover, a gain of 27 dB in the $E_{b}/N_{0}$ was achieved through the employment of the source bit coding in conjunction with rate-1/3 SBC against the benchmarker.

## Figures and Tables

**Figure 1 sensors-21-05461-f001:**
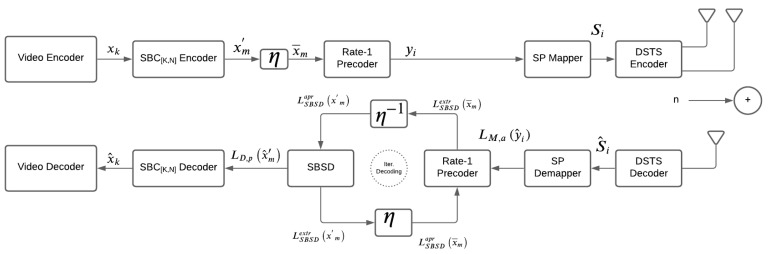
System model.

**Figure 2 sensors-21-05461-f002:**
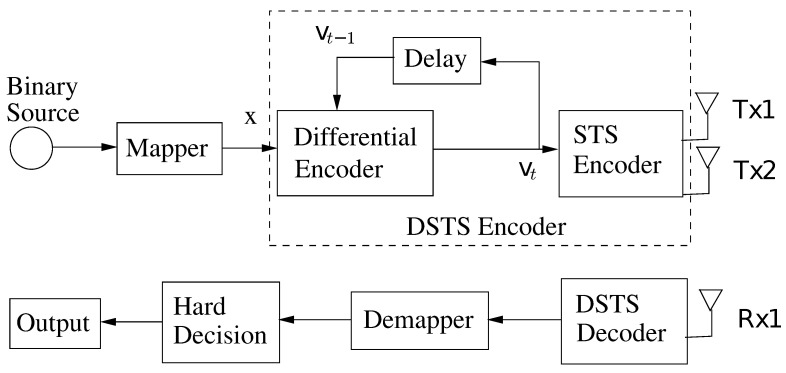
DSTS encoder and decoder.

**Figure 3 sensors-21-05461-f003:**
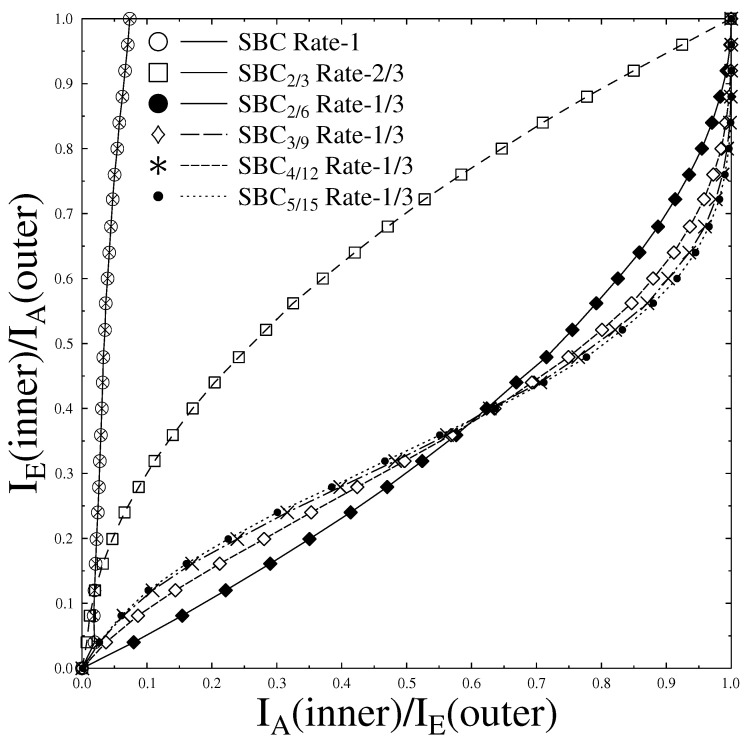
EXIT Chart analysis with varying code rates.

**Figure 4 sensors-21-05461-f004:**
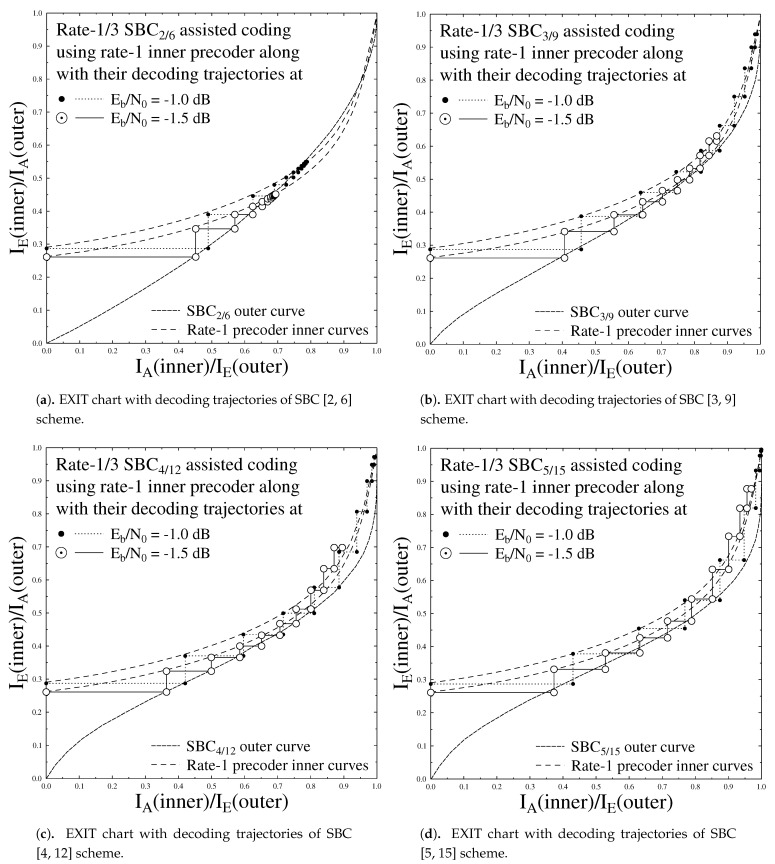
EXIT charts with decoding trajectories.

**Figure 5 sensors-21-05461-f005:**
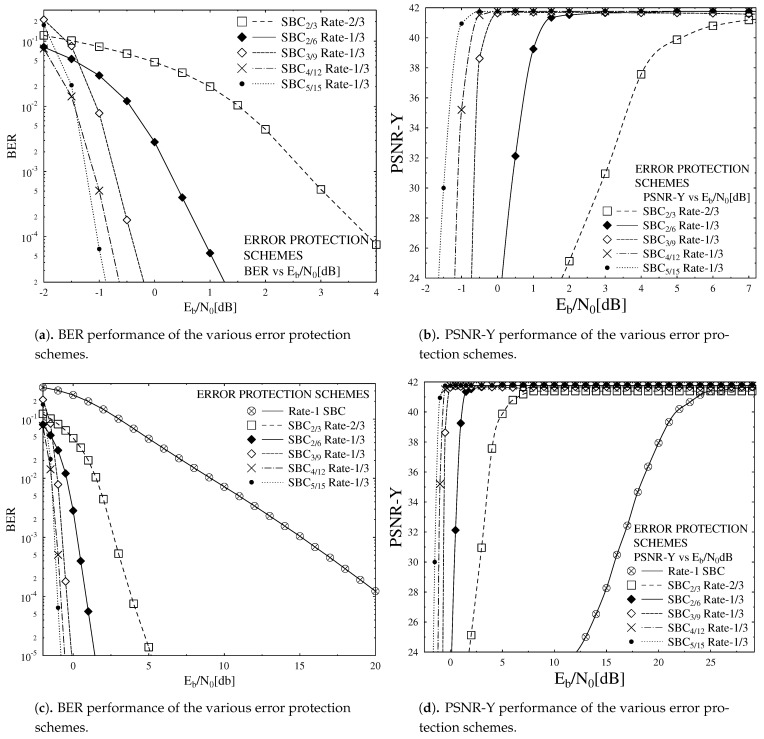
BER and PSNR performance of the proposed system.

**Table 1 sensors-21-05461-t001:** Different SBCs with corresponding symbols and $d_{H,min}$.

SBC Type	Symbol in Decimal	$d_{H,\min}$
Rate-1 SBC	{0,1}	1
Rate-2/3 SBC(2/3)	{0,3,5,6}	2
Rate-1/3 SBC(2/6)	{0,22,41,63}	3
Rate-1/3 SBC(3/9)	{0,78,149,219,291,365,438,504}	4
Rate-1/3 SBC(4/12)	{0,286,557,819,1099,1365,1638,1912,2183, 2457,2730,2996,3276,3538,3809,4095}	5
Rate-1/3 SBC(5/15)	{0,1086,2141,3171,4251,5285,6342,7416, 8471, 9513, 10,570, 11,636, 12,684, 13,746, 14,801, 15,855, 16,911, 17,969, 19,026, 20,076, 21,140, 22,186, 23,241, 24,311, 25,368, 26,406, 27,461, 28,539, 29,571, 30,653, 31,710, 32,736}	6

**Table 2 sensors-21-05461-t002:** Code rates for different error protection schemes.

Error Protrction Scheme	Code Rate		
	Outer Code (SBC)	Inner Code	Overall
SBC Rate-1	Rate-1	Rate-1 Precoder	1/3
SBC-2/3	Rate-2/3	Rate-1/2 SBC	1/3
SBC-2/6	Rate-1/3	Rate-1 Precoder	1/3
SBC-3/9	Rate-1/3	Rate-1 Precoder	1/3
SBC-4/12	Rate-1/3	Rate-1 Precoder	1/3
SBC-5/15	Rate-1/3	Rate-1 Precoder	1/3

**Table 3 sensors-21-05461-t003:** Systems  parameters.

Parameters	Value	Systems Parameters	Value
Source Coding	H.264/AVC	Modulation Scheme	SP
Bit Rate (kbps)	64	MIMO Scheme	DSTS
Frame Rate (fps)	15	Number of transmitters	2
No of Slices/frame	9	Number of receivers	1
Intra-frame MB update/frame	3	Number of Users	4
Channel Coding	Rate-1 Precoder	Channel	Correlated Rayleigh fading
Over-all Code Rate	1/3	Normalized Doppler Frequency	0.01

## Data Availability

Not Applicable.

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
