# Peer review of "Performance Analysis of Sphere Packed Aided Differential Space-Time Spreading with Iterative Source-Channel Detection"

_sensors, 2021, doi:10.3390/s21165461_

Round 1
Reviewer 1 Report
The paper presents novel approach of using Sphere Packing (SP) Modulation aided Differential Space Time Spreading (DSTS). However, there are some concerns especially with the presentation of the article.
- 5G communication is used in Title and in first line of abstract only. There is no discussion on how the proposal is related to 5G.
- What is SBSD? What is ISCD? Elaborate SBCs when first writing in Introduction.
- In related work section write few sentences to show how the proposed work is different from existing works.
- Introduction can be improved. It does not give clear picture of what the article is about. Motivation and Contribution should be highlighted in Introduction. Section 3 on Motivation and Contribution is too late and too little.
- While discussing motivation, more details can be provided on fast fading channel and related problem.
- Authors claim in Section 3 that different code rates of the channel coding techniques have varying effects on the
results. Please provide couple of more sentences to explain the same. - Why suddenly Section 4 discuss H.264? There is no continuity. Please relate it clearly to the proposed work.
- Why do authors consider the transmission channels used for DSTS to be correlated?
- How do results relate to 5G system?
Reviewer 2 Report
This paper is entitled “EXIT Chart Analysis of Sphere Packed Symbols Assisted Differential Space Time Spreading Optimized Iterative Detection for Robust 5G Communication” which already is a huge title. And it appears to me that the whole set of ideas is also not well explained in the overall paper, which is not easy to read.
Partially, this is because the paper has many English typos and errors, so an overall revision namely by some person that is English fluent will help and it is, in my view, mandatory. And not only the errors, but as I said, the ideas are very confuse.
I can point out some things but not exhaustively as the paper, in my opinion, needs to be revised. Just to mention a few:
Page 1-line 1: Why do you think 5G has excessively high speeds?
Page 1-line 5: setup and not “set up”
Page 1-line 8: detection “are” used…
Page 1-line 16: “outperforms”
Page 2-line 63: Why motivation comes only in Section 3? It should be in the introduction!
Page 5-equation 1: curve brackets missing in the numerator
Page 9-Table 3: Bad designed table. At least the middle line should be different as it seems a table with 9 rows and 4 columns and in fact it is a table with 17 rows and 2 columns.
Page 9-Figure 3: this is not a system model as the label says…
Page 10-line 272: it is also strange that a system needs 10 iterations… That is a very slow convergence.
Page 12-line 283: claiming a gain of 27 dB seems to me quite excessive for a real system…
Concluding I think that this paper needs a throughout revision before it can be resubmitted to Sensors.
Author Response
Please see the attachment for the comments
Thank you

Round 2
Reviewer 1 Report
- H.264 is a widely used video encoder. the same should be clearly mentioned and little more explanation should be given on this. Section 3 still looks like incomplete.
- Some minor changes are required a. M in multimedia could be small in first sentence of Introduction. b. Avoid using however again again like in sentence 70. c. Why 'W" is capital in sentence 70. d. 'more training sequence needs to be sent.' etc. Please proof read.
- Future multimedia systems would require to support Ultra-High Definition (UHD) video transmission. A small paragraph on this and 5G would add value to manuscript.
Author Response
Thank you for providing valuable suggestions. Here point-by-point response is as:
Reviewer Suggestion 1: H.264 is a widely used video encoder. the same should be clearly mentioned and little more explanation should be given on this. Section 3 still looks like incomplete.
Response: We agree with this suggestion and have provided more details about the H.264 encoder and highlighted it in the updated manuscript.
Reviewer Suggestion 2: Some minor changes are required a. M in multimedia could be small in first sentence of Introduction. b. Avoid using however again again like in sentence 70. c. Why 'W" is capital in sentence 70. d. 'more training sequence needs to be sent.' etc. Please proof read.
Response: Thank you for highlighting this. We have removed the grammatical mistakes after proofreading.
Reviewer Suggestion 3: Future multimedia systems would require to support Ultra-High Definition (UHD) video transmission. A small paragraph on this and 5G would add value to manuscript.
Response: Thank you for highlighting this. We have provided a paragraph about 5G and highlighted it in the updated manuscript.